# Elevated Allele Frequency and Male-Predominance of a Common *LAG3* Germline Variant in Multiple Myeloma

**DOI:** 10.3390/cimb48010005

**Published:** 2025-12-20

**Authors:** Katja Seipel, Alina Mena, Pinar Horum, Michele Hoffmann, Inna Shaforostova, Ulrike Bacher, Thomas Pabst

**Affiliations:** 1Department of Medical Oncology, Inselspital, University Hospital Bern, University of Bern, 3010 Bern, Switzerland; alina.mena@students.unibe.ch (A.M.); pinar.horum@students.unibe.ch (P.H.); michele.hoffmann@insel.ch (M.H.);; 2Department for Biomedical Research, University of Bern, 3008 Bern, Switzerland; 3Department of Hematology, University Hospital Bern, 3010 Bern, Switzerland; veraulrike.bacher@insel.ch

**Keywords:** lymphocyte activation gene 3 (*LAG3*), cytotoxic T-lymphocyte associated protein 4 (CTLA4), autologous stem cell transplantation (ASCT), single nucleotide polymorphism (SNP), minor allele frequency (MAF), multiple myeloma (MM), genetic risk

## Abstract

The incidence of multiple myeloma is higher in males. The underlying mechanisms may be related to differences in immune system orchestration in males and females. LAG3 and CTLA4 are immune checkpoint proteins and inhibitory regulators of T cells. Here, we analyzed the prevalence of the common *LAG3* gene variant rs870849 and the common *CTLA4* gene variant rs231775 in myeloma patients eligible for autologous stem cell transplantation. *CTLA4* rs231775 was prevalent at normal allele frequencies. In contrast, *LAG3* rs870849 was prevalent at elevated allele frequencies in myeloma patients, with allele frequency 0.61 in male and 0.53 in female patients compared to 0.39 in the European population. The gene risk analysis of rs870849 indicated an odds ratio 6.8 in male and 3.6 in female patients. Moreover, treatment outcomes differed in the three genetic *LAG3* subgroups with median progression-free survival of 2.6, 3.3 and 3.4 and median overall survival of 7, 15 and 18 years in the I455hom, I455Thet and T455hom subgroups, respectively. *LAG3* rs870849 may affect survival and treatment outcome after autologous stem cell transplantation in myeloma patients with favorable outcomes in rs870849 carriers.

## 1. Introduction

The incidence of hematological malignancies increases significantly with age and the incidence rates are generally higher in men than women [1]. Male predominance is thought to be due to a combination of factors, including environmental exposures and hormonal and immune system differences [2,3]. Multiple myeloma (MM) is a plasma cell malignancy characterized by clonal proliferation of terminally differentiated B cells in the bone marrow, leading to monoclonal protein production, bone destruction and immunodeficiency [4]. Despite major therapeutic advances MM remains largely incurable [5]. Treatment resistance has been linked to clonal evolution and support from the immunosuppressive bone marrow microenvironment [6,7].

Extensive real-world cohorts have confirmed heterogeneous outcomes and poor long-term survival in some transplant-eligible MM patients, emphasizing the need for improved risk stratification and molecularly guided therapies [8,9]. Within the bone marrow microenvironment, malignant plasma cells create conditions that impair antitumor immunity through effector T cell exhaustion and expansion of regulatory T cells [7]. Several immune checkpoint molecules—such as programmed death-1 (PD-1), cytotoxic T-lymphocyte-associated protein 4 (CTLA-4, CD152) and lymphocyte activation gene 3 (*LAG3*, CD223)—play central roles in orchestrating these immune interactions [10]. Among them, LAG3 has emerged as a key regulator of T cell exhaustion and a promising target in next-generation tumor immunotherapy [10,11].

The *LAG3* gene, located on chromosome 12p13.31, encodes a type I transmembrane protein expressed on activated CD4+ and CD8+ T cells, natural killer cells and dendritic cells [11]. LAG3 binds to major histocompatibility complex class II (MHC-II) and to fibrinogen-like protein 1 (FGL1), transmitting inhibitory signals that reduce T cell proliferation and cytokine secretion [12]. Overexpression of LAG3 on T cells has been observed in patients with MM and correlates with disease activity and diminished immune surveillance [13,14]. In preclinical models, LAG3 blockade restores T cell cytotoxicity and enhances anti-myeloma responses, particularly when combined with PD-1 inhibition [15]. These findings suggest that LAG3 contributes to T cell exhaustion and disease progression in MM [13,15].

*LAG3* gene variants may have differential impact on immune checkpoint signaling. *LAG3* gene single-nucleotide polymorphisms (SNPs) can alter transcription, splicing and receptor conformation, thereby modifying immune tolerance thresholds [16]. The common germline missense variant rs870849 (C>T; I455T) and the intron variant rs2365095 (A>G) have been linked to immune dysregulation and variable disease susceptibility [17]. Genetic polymorphisms in immune-regulatory genes such as *CD19*, *CTLA4* and *LAG3* have been reported to affect treatment response in hematological malignancies treated with CAR-T cell therapy [18,19,20]. These findings indicate that germline variation in immune-related genes may affect immune modulation and therapy responses. Donor *LAG3* rs870849 genotypes have been linked to adverse outcomes following allogeneic stem cell transplantation, while other *LAG3* polymorphisms have been associated with altered immune recovery in bone marrow failure syndromes [16,17]. LAG3 acts in concert with other inhibitory receptors—including PD-1, TIM-3 and TIGIT—to form co-inhibitory networks that tightly regulate immune activation and tolerance, as reviewed in [21,22]. In this context, germline *LAG3* variation may represent a stable genetic factor predisposing to inadequate immune surveillance and tumor persistence. The broad clinical application of combined PD-1/LAG3 blockade underscores the translational relevance of this pathway [23]. Understanding the genetic landscape of LAG3 polymorphisms may thus provide novel insights into MM pathogenesis and support the development of individualized immunotherapeutic strategies.

In this study, we analyzed the potential gene risk of the common *LAG3* gene variant rs870849 in the emergence of MM and differences in treatment efficacies depending on the *LAG3* genetic background. By genotyping well-defined patient and control cohorts and correlating allelic variants with clinical parameters, we aimed to clarify the contribution of *LAG3* genetic variation to MM susceptibility and treatment response. Identification of risk-associated *LAG3* variants may elucidate disease mechanisms and improve personalized immunotherapy strategies in MM.

## 2. Materials and Methods

### 2.1. Gene Sequence and Risk Analysis

The *CTLA4* and *LAG3* genes were analyzed by Sanger sequencing as previously described [18,19]. Gene risk was analyzed on a Hardy–Weinberg equilibrium calculator [24].

### 2.2. Clinical Data

We conducted a retrospective single-center study at the Inselspital, University Hospital Bern, Switzerland. The patient cohort comprised 171 patients diagnosed with multiple myeloma in the two decades from July 2003 to June 2022. All 171 patients received induction therapy, 156 patients received HDCT/ASCT. The study was performed following the Declaration of Helsinki and approved by the Ethics Commission of the Canton of Bern (decision number 2025-00853, date of approval 24 April 2025). All patients signed a written informed consent form. Clinical Data were analyzed as previously described [25].

## 3. Results

### 3.1. Elevated Allele Frequency of LAG3 rs870849 in Patients with Multiple Myeloma

The sequence of the *CTLA4* gene exon one and the *LAG3* gene exon seven were determined in the peripheral blood cells of 171 MM patients. In total, 72 patients (42%) carried two major *CTLA4* alleles encoding CTLA4 T17 (T17hom), 74 patients (43%) had one allele rs231775 (T17Ahet) and 23 patients (14%) carried two alleles rs231775 (A17hom). Further, 25 patients (15%) carried two major *LAG3* alleles encoding LAG3 I455 (I455hom), 91 patients (53%) had one allele rs870849 (I455Thet) and 55 patients (32%) carried two alleles rs870849 (T455hom). Allele frequencies and genetic risks were analyzed for the MM cohort on a Hardy–Weinberg equilibrium calculator [24], compared to allele frequencies in the European population with *CTLA4* rs231775 and *LAG3* rs870849 allele frequencies of 0.36 and 0.39, respectively (ALFA sample size 588,090, https://www.ncbi.nlm.nih.gov/snp/, access date 22 August 2025). The observed *CTLA4* rs231775 allele frequency 0.36 in the MM group was identical to the European control group (Figure 1A–C), while the *LAG3* rs870849 allele frequency 0.59 was significantly higher in the MM group than in the European control group with rs870849 allele frequency 0.39 (Figure 1D–F). The MM cohort consisted of 110 (64%) males and 61 (36%) females, an m/f ratio of 1.8 (Table 1). The *LAG3* rs870849 allele frequency was 0.54 in MM female and 0.61 in MM male groups (Figure 2A,B). Males predominated within the T455hom group with m/f ratio of 2.24 (71% males), while the m/f ratio was 1.84 (65% males) in the I455Thet and 0.93 (48% males) in the I455hom group, respectively (Figure 2C,D).

### 3.2. Gene Risk Analysis

Gene risk associated with *LAG3* rs870849 and *CTLA4* rs231775 variants was analyzed on Hardy–Weinberg equilibrium calculator in different genetic models including dominant, codominant and recessive [24]. In all genetic models *LAG3* rs870849 was significantly associated with risk for multiple myeloma (Figure 3A). LAG3 protein variants may form heterodimers and all combinatorial effects are possible including dominant, codominant and recessive dimer function. The gene risk associated with *LAG3* rs870849 4 in the codominant model was dose-dependent with OR 5.4 at dosage 2 vs. 0, OR 2.9 at dosage 1 vs. 0 and OR 1.9 at dosage 2 vs. 1 (Figure 3B). In the stratified analysis male-specific risk was higher with OR 6.8 compared to female-specific risk OR 3.6 (Figure 3C). In contrast, *CTLA4* rs231775 was not associated with risk for MM with an OR of 1.1, with male-specific OR 1.1 and female-specific OR 0.9 (Figure 3D–F).

**Figure 1 cimb-48-00005-f001:**
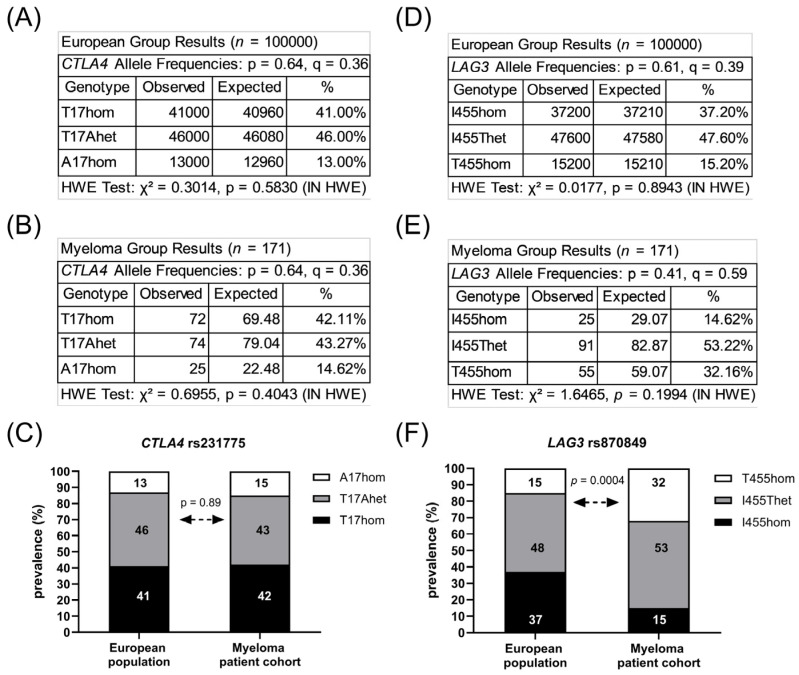
Prevalence of *CTLA4* rs231775 and *LAG3* rs870849. Allele frequencies of *CTLA4* rs231775 in the European population (**A**) and in the local MM cohort (**B**). Prevalence of *CTLA4* rs231775 in the local MM cohort compared to the European population (**C**). Allele frequencies of *LAG3* rs870489 in the European population (**D**) and in the local MM cohort (**E**). Prevalence of *LAG3* rs870849 in the local MM cohort compared to the European population (**F**). HWE: Hardy–Weinberg equilibrium. p: major allele, q: minor allele.

**Figure 2 cimb-48-00005-f002:**
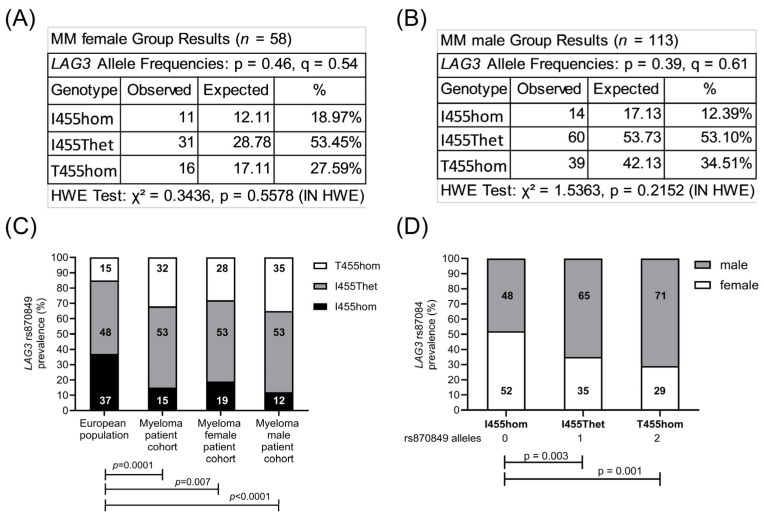
Male predominance of *LAG3* rs870849 in the MM cohort. Allele frequencies were analyzed according to the *LAG3* genetic variants in the MM female (**A**) and MM male (**B**) subgroups. Prevalence of *LAG3* rs870849 in the MM subgroups compared to the European population (**C**) and gender distribution in the *LAG3* genetic subgroups of myeloma patients (**D**). HWE: Hardy–Weinberg equilibrium. p: major allele, q: minor allele.

**Figure 3 cimb-48-00005-f003:**
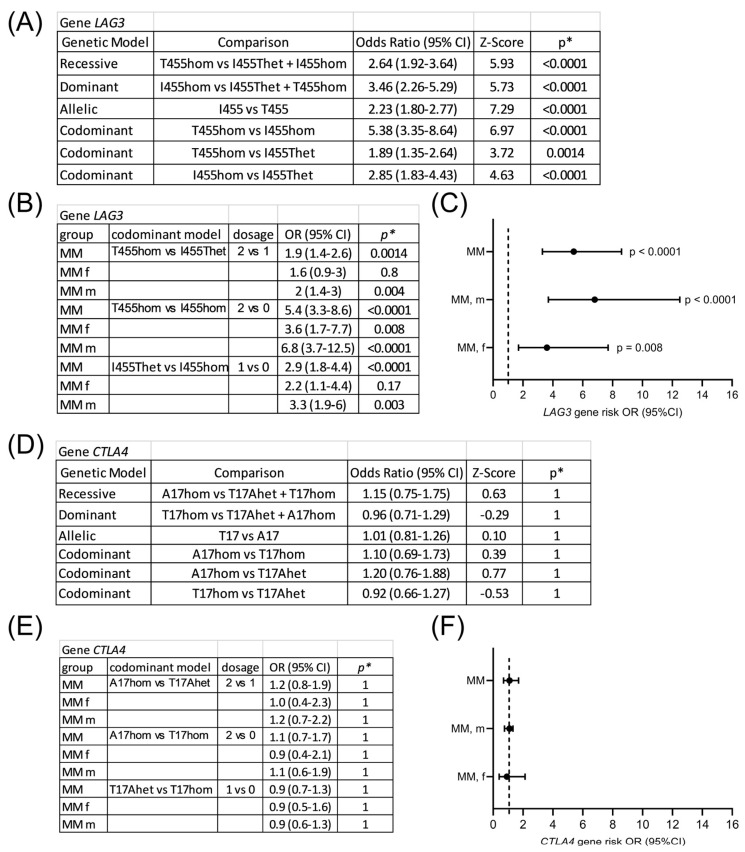
Gene risk associated with *LAG3* rs870849 and *CTLA4* rs231775 in multiple myeloma. *LAG3* gene risk analysis in different genetic models (**A**) and in codominant models, stratified for males and females (**B**). Forest plot of gene risk associated with *LAG3* T455hom vs. I455hom (**C**). *CTLA4* gene risk analysis (**D**,**E**). Forest plot of gene risk associated with *CTLA4* A17hom vs. T17hom (**F**). OR: odds ratio; CI: confidence interval. * *p* values with Bonferroni correction for multiple testing (α = 0.05/7 = 0.00714).

### 3.3. Clinical Characteristics in the Myeloma Cohort According to LAG3 rs870849

The MM cohort comprised 171 patients diagnosed with MM at a median age of 60 years, including 110 males and 61 females, indicating a sex ratio (m/f) of 1.8. Baseline clinical characteristics were analyzed for the entire cohort and for the three genetic subgroups with LAG3 rs870849 encoding isoleucine or threonine at the amino acid position of the LAG-3 protein (LAG3 I455hom, I455Thet, T455hom) (Table 1). The majority of patients were characterized by male gender, ISS II, standard cytogenetic risk, paraprotein types IgG and light chain kappa, normal hemoglobin, calcium and LDH levels, elevated beta-2-microglobulin levels, as well as high bone marrow infiltration rates. Half of the patients were anemic (Hb < 110 g/L) and had decreased serum albumin levels, with renal dysfunction observed in 40% of the patients. The clinical characteristics were comparable in the three genetic subgroups with the exception of the median age at diagnosis and the sex ratio; the median age at the time of initial diagnosis was 56 years in the I455hom and 60 years in the I455Thet and T455hom subgroups (*p* = 0.07). Male predominance was elevated in the T455hom subgroup with a sex ratio (m/f) of 2.44 (*p* = 0.02) and reduced in the I455hom subgroup with a sex ratio (m/f) of 0.92. Paraprotein composition differed in the three genetic subgroups: while IgG paraprotein prevailed in the I455hom and I455Thet subgroups (*p* = 0.03), the T455hom subgroup had the highest proportion of light chain only paraprotein (*p* = 0.05) and renal dysfunction (*p* = 0.04). Anemia and hypercalcemia prevailed in the *LAG3* r2870849 carriers (T455hom and I455Thet).
cimb-48-00005-t001_Table 1Table 1Clinical characteristics according to *LAG3* genotype.
All Patients(N = 171)I455hom(N = 25)I455Thet(N = 91)T455hom(N = 55)*p*-ValueSex ratio (m/f)1.80.921.842.44 0.02 Female, *n* (%)61 (36%)13 (52%)32 (35%)16 (29%)0.14Male, *n* (%)110 (64%)12 (48%)59 (65%)39 (71%)Age at ID, median (range)60 (33–78)56 (42–73)60 (39–78)60 (33–74)0.07Initial disease stage (ISS)N = 159N = 23N = 85N = 510.59I50 (31%)9 (39%)25 (30%)16 (31%)0.67II59 (38%)10 (44%)30 (35%)19 (38%)0.74III50 (31%)4 (17%)30 (35%)16 (31%)0.19Cytogenetic risk categoryN = 120N = 18N = 62N = 400.36high risk *34 (28%)6 (33%)20 (32%)8 (20%) standard risk86 (72%)12 (67%)42 (68%)32 (80%) Cytogenetic aberrationsN = 120N = 18N = 62N = 400.85−1334 (28%)6 (33%)21 (34%)7 (18%)0.17−148 (7%)1 (6%)4 (6%)3 (8%)0.99−165 (4%)2 (11%)3 (5%)00.11+1 q41 (34%)8 (44%)19 (31%)14 (35%)0.54del17p *13 (11%)1 (6%)8 (13%)4 (10%)0.78t(4;14) *19 (16%)3 (17%)12 (19%)4 (10%)0.48t(14;16) *8 (7%)2 (11%)5 (8%)1 (3%)0.31t(11;14)27 (23%)3 (17%)16 (26%)8 (20%)0.72Paraprotein-type    0.18Heavy chain IgG111 (65%)19 (76%)64 (70%)28 (51%)0.03Heavy chain IgA33 (19%)4 (16%)16 (17%)13 (24%)0.64Light chain only26 (15%)2 (8%)10 (12%)14 (25%)0.05Light chain kappa112 (66%)15 (60%)60 (66%)37 (67%)0.86Light chain lambda59 (34%)10 (40%)31 (34%)18 (33%)0.86AnemiaN = 143N = 23N = 75N = 45 Hb (g/L), median (range)109 (46–167)114 (70–148)109 (46–164)106 (71–167)0.87Hb < 110 g/L, *n* (%)72 (51%)8 (35%)38 (51%)26 (58%)0.21HypercalcemiaN = 117N = 18N = 62N = 37 Ca (mmol/L), median (range)2.42 (1.4–4.2)2.35 (2.1–2.8)2.45 (1.4–4.2)2.43 (2.1–4.2)0.13>2.6 mmol/L, *n* (%)24 (21%)1 (6%)14 (23%)9 (24%)0.24Beta-2-microglobulinN = 143N = 23N = 74N = 46 B2M (mg/L), median (range)3.3 (1.18–38)3.2 (1.8–7.3)3.4 (1.18–30)3.1 (1.51–38)0.77>2.2 mg/L, *n* (%)104 (73%)20 (87%)54 (73%)30 (65%)0.16Lactate-dehydrogenaseN = 103N = 21N = 49N = 33 LDH (U/L), median (range)274 (110–1366)272 (162–644)276 (110–1366)259 (145–526)0.46>480 U/L, *n* (%)11 (11%)2 (10%)5 (10%)4 (12%)0.99Serum albuminN = 133N = 22N = 70N = 41 g/dL, median (range)3.6 (1.8–29)3.6 (1.8–8)3.4 (1.8–29)3.6 (2.5–5)0.45<3.5 g/dL, *n* (%)62 (47%)7 (32%)35 (50%)14 (34%)0.16Bone marrow infiltrationN = 167N = 25N = 89N = 53 Percent, median (range)60 (3–100)60 (10–90)70 (5–100)55 (3–100)0.3740–100%, *n* (%)122 (73%)19 (76%)68 (76%)35 (66%)0.39Osteolytic lesions122 (74%)16 (67%)65 (73%)41 (79%)0.52Renal dysfunction51 (40%)6 (33%)26 (37%)20 (50%)0.04ISS: international staging system; LDH: lactate dehydrogenase. * High risk cytogenetics include del17p, t(4;14), t(14;16).

### 3.4. Clinical Response to HDCT/ASCT According to LAG3 rs870849

All 171 patients diagnosed with MM received induction therapy, consisting of proteaseome inhibitor bortezomib (velcade) and dexamethasone in combination with lenalidomide (revlimid) or cyclophosphamide (Table 2). More recently, the monoclonal antibody daratumumab was added to induction therapy (Dara-VRD). Clinical response to induction therapy included 11–13% complete response (CR) and 58–62% partial response (PR) in the I455Thet and T455hom subgroups, with 88% PR in the I455hom subgroup. Relapsed patients were treated with immuno-chemotherapy (ICT). Patients with multiple relapses received multiple ICT lines. A total of 156 patients received high-dose chemo-therapy (HDCT) followed by autologous stem cell transplant (ASCT). Clinical responses to HDCT/ASCT differed in the three genetic subgroups with a median progression-free survival (PFS) of 2.6, 3.3 and 3.4 years and a median overall survival (OS) of 7, 15 and 18 years in the I455hom, I455Thet and T455hom subgroups, respectively (Figure 4). In the multivariate analysis, the *LAG3* variant was not associated with survival outcome, with initial disease stage, high risk cytogenetics and age as confounding predictors (Table 3).

## 4. Discussion

Risk factors for MM include age, sex and ethnicity. The median age at initial diagnosis of MM is 70 years. Men are more likely to develop MM than women. The incidence of MM is higher in people of African descent than in Caucasians. In our analysis of a European MM patient cohort, we defined a male-predominant risk allele for MM, a common human *LAG3* germline variant. *LAG3* rs870849 was prevalent at elevated allele frequencies in MM patients eligible for ASCT, with MAF 0.61 in male and MAF 0.53 in female MM patients compared to MAF 0.39 in the general European population, and MAF 0.45 in the African and African American population. In all genetic models, *LAG3* rs870849 was a significant risk allele, in the codominant model at gene dosage 2 vs. 0 with an OR of 5.4, stratified into male-specific OR 6.8 and female-specific OR 3.6.

Sex-specific risk loci have been defined for other medical conditions with sex-specific differences in prevalence including asthma, coronary artery disease and colorectal cancer [26,27,28]. Genome-wide association studies (GWAS) have identified common risk alleles for MM at 24 independent loci [29]. Other germline risk alleles may contribute to genomic instability and MM [30]. Risk alleles for MM were also defined in genes related to activation, detoxification and clearance of chemical carcinogens [31]. *LAG3* rs870849 on 12p13 may be a significant male-predominant myeloma risk allele, possibly with a bias toward light chain myeloma with renal dysfunction.

*LAG3* rs870849 was previously studied in DLBCL patients treated with CAR-T cell therapy where it was present at elevated frequencies and implicated in favorable treatment outcome [19]. In the current study the germline allele *LAG3* rs870849 was detected at elevated frequencies in MM patients eligible for HDCT/ASCT and associated with favorable treatment outcome. The effects of *LAG3* rs870849 were dose-dependent, indicating functional differences in the LAG3 protein variants due to the substitution of isoleucine to threonine in the LAG3 transmembrane domain (I455T). The molecular mechanisms underlying the opposing traits of promoting and impeding the same disease may be related to the appropriate balance of T cell activity and T cell exhaustion [32,33,34]. In the promotion of myeloma development, *LAG3* rs870849 may increase the number of exhausted T cells with concomitant loss of immune surveillance. In stem cell-based therapy *LAG3* rs870849 may promote T cell activity against myeloma cells.

Limitations of the study include the cohort size and composition with a large majority of patients who underwent HDCT/ASCT (90%). The *LAG3* rs870849 allele frequencies were found to be elevated in MM patients eligible for ASCT but may be lower in MM patients not eligible for ASCT. This may represent a survivorship bias with the implication that *LAG3* rs870849 may confer a survival advantage. Most patients diagnosed with MM are at an advanced age and eligibility for HDCT/ASCT has been limited. Traditionally, a cutoff of 65 years was implemented, but this is shifting to 75 years, with recent data showing improved outcomes for older patients undergoing ASCT [35]. However, the commonly applied induction therapy may impair the stem cells in MM patients leading to a reduced reconstitution potential after ASCT [36]. In stem cell-based therapies *LAG3* rs870849 may promote stem cell function and reconstitution potential. The molecular mechanisms involving different LAG3 variants remain to be established in future studies.

## Figures and Tables

**Figure 4 cimb-48-00005-f004:**
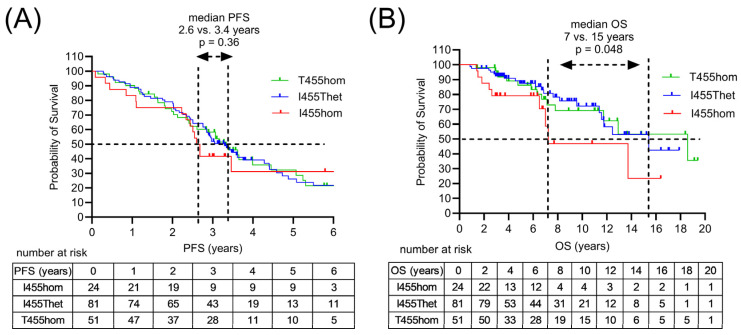
Survival outcomes of MM patients following HDCT/ASCT according to the *LAG3* genotype. (**A**) Progression-free survival (PFS) according to *LAG3* gene polymorphism rs870849 encoding LAG3 I455hom, I455Thet or T455hom. (**B**) Overall survival (OS) in the genetic subgroups LAG3 I455hom, I455Thet and T455hom.

**Table 2 cimb-48-00005-t002:** Therapy response according to *LAG3* genotype.

	All Patients (N = 171)	I455hom (N = 25)	I455Thet (N = 91)	T455hom (N = 55)	*p*-Value
Induction therapy					0.24
VRD	70 (41%)	13 (52%)	39 (43%)	18 (33%)	
VCD	45 (27%)	7 (28%)	24 (26%)	14 (25%)	
VD	18 (10%)	2 (8%)	10 (11%)	6 (11%)	
Dara-VRD	18 (10%)	3 (12%)	10 (11%)	5 (9%)	
Other	20 (12%)	0	8 (9%)	12 (22%)	
Response to induction therapy					0.22
CR	18 (10%)	0	12 (13%)	6 (11%)	
PR	109 (64%)	22 (88%)	53 (58%)	34 (62%)	
SD/PD	8 (5%)	0	5 (6%)	3 (5%)	
Not reported	36 (21%)	3 (12%)	21 (23%)	12 (22%)	
Number of Relapses, *n* (%)					0.32
0	46 (27%)	8 (32%)	22 (24%)	16 (29%)	
1–2	49 (29%)	6 (24%)	29 (32%)	14 (25%)	
3–4	43 (25%)	4 (16%)	26 (29%)	13 (24%)	
5–8	33 (19%)	7 (28%)	14 (15%)	12 (22%)	
Response to HDCT/ASCT	N = 156	N = 22	N = 91	N = 43	0.82
CR	92 (59%)	12 (55%)	52 (57%)	28 (65%)	
PR	31 (20%)	5 (23%)	17 (19%)	9 (21%)	
SD/PD	2 (1%)	0	2 (2%)	0	
Not reported	31 (20%)	5 (23%)	20 (22%)	6 (14%)	
Radiotherapy	69 (40%)	7 (28%)	39 (43%)	23 (42%)	0.41
PFS, years, median	3.1	2.6	3.3	3.4	0.36
OS, years, median	14	7	15	18	0.048

VRD: Velcade, Revlimid, Dexamethasone; VCD: Velcade, Cyclophosphamide, Dexamethasone; VD: Velcade, Dexamethasone; CR: complete response; PR: partial response; SD: stable disease; PD: progressive disease; HDCT: high-dose chemotherapy; ASCT: autologous stem cell transplantation.

**Table 3 cimb-48-00005-t003:** Clinical outcome hazard ratios (HR), multivariate analysis.

	PFS		OS	
Predictors	HR	*p*	HR	*p*
*LAG3* I455 hom vs. I455Thet	0.98	0.97	0.92	0.90
*LAG3* I455hom vs. T455hom	1.19	0.67	0.97	0.97
ISS3 vs. ISS1	1.30	0.31	1.39	0.47
Cytogenetic risk high vs. standard	1.35	0.43	2.41	0.14
Age at diagnosis >60 y vs. <60 y	0.90	0.65	2.41	0.06

PFS: progression-free survival, OS: overall survival; ISS: International Staging System.

## Data Availability

The original contributions presented in this study are included in the article. Further inquiries can be directed to the corresponding authors.

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
