# Peer review of "Elevated Allele Frequency and Male-Predominance of a Common LAG3 Germline Variant in Multiple Myeloma"

_cimb, 2025, doi:10.3390/cimb48010005_

Round 1

Reviewer 1 Report

Comments and Suggestions for Authors

Major Comments

1. A multivariate analysis of overall survival (OS) is essential to determine whether the LAG3 genotype is an independent prognostic factor, unrelated to age, ISS stage, high-risk cytogenetics, or paraprotein type.
2. The study cohort consists exclusively of patients eligible for HDCT/ASCT, which may introduce selection bias. This could lead to an overestimation of the frequency of the LAG3 risk allele compared with the broader multiple myeloma (MM) population.
3. The authors should be more cautious when drawing conclusions about the proposed mechanisms of the LAG3 allele (risk vs. improved treatment response), as no experimental evidence is provided to support these hypotheses.

Minor Comments
4.It would be valuable to expand the discussion on how the LAG3 genotype relates to specific clinical features, such as light-chain–only paraproteinemia and renal dysfunction.
5. Please ensure that the abbreviations (e.g., HW, HT, HR) are used consistently and clearly defined throughout the manuscript.

Author Response

Reviewer 1 cimb-4029985

Response: We thank the reviewer for the helpful comments. We integrated the responses into the article and believe that the manuscript has been improved.

Major Comments
1. A multivariate analysis of overall survival (OS) is essential to determine whether the LAG3 genotype is an independent prognostic factor, unrelated to age, ISS stage, high-risk cytogenetics, or paraprotein type.

Response: In the multivariate analysis, the LAG3 variant was not associated to survival outcome, with initial disease stage, high risk cytogenetics and age as confounding predictors (Table 3). 

  1. The study cohort consists exclusively of patients eligible for HDCT/ASCT, which may introduce selection bias. This could lead to an overestimation of the frequency of the LAG3 risk allele compared with the broader multiple myeloma (MM) population.

Response: We had stated in the original discussion, that the main limitation of the study was the cohort composition with a large majority of patients who underwent HDCT/ASCT (90%). We have elaborated on the topic by discussing the potential survivorship bias and possible implications. The LAG3 rs870849 allele frequencies were found to be elevated in MM patients eligible for ASCT, but may be lower in MM patients not eligible for ASCT. This may represent a survivorship bias with the implication that LAG3 rs870849 may confer a survival advantage.

  1. The authors should be more cautious when drawing conclusions about the proposed mechanisms of the LAG3 allele (risk vs. improved treatment response), as no experimental evidence is provided to support these hypotheses.

Response: We have removed the term risk allele from the title.

Minor Comments
4. It would be valuable to expand the discussion on how the LAG3 genotype relates to specific clinical features, such as light-chain–only paraproteinemia and renal dysfunction.

Response: We have expanded the discussion to specify the association of LAG3 rs870849 to light chain myeloma with renal dysfunction. LAG3 rs870849 on 12p13 may be a significant male-predominant myeloma risk allele, possibly with a bias toward light chain myeloma with renal dysfunction.

  1. Please ensure that the abbreviations (e.g., HW, HT, HR) are used consistently and clearly defined throughout the manuscript.

Response: We have addressed the definitions and consistency of abbreviations.

Submission Date 21 November 2025

Date of this review 06 Dec 2025 10:05:36

Date of revision 15 December 2025.

Reviewer 2 Report

Comments and Suggestions for Authors

In this study by Seipel et al., the authors conduct a retrospective analysis of allele frequency of the LAG3rs870849 variant and its association with gender and clinical outcomes in patients with multiple myeloma. The authors show that the homozygous LAG3 rs870849 variant is predominant in multiple myeloma patients and is also more prevalent in male patients than in female patients. Interestingly, patients with the homozygous LAG3rs870849 variant had significantly longer median overall survival.

The manuscript is well written and observations from the study emphasize the need for better characterization of the molecular mechanisms involved in LAG3 rs870849 variant modulation of T-cell activity in different contexts that might contribute to disease risk vs. response to treatment.

There are some concerns that the authors must address.

  • It is better to avoid abbreviations in the Abstract. All abbreviations must be expanded in the first instance in the text.
  • “Genetic polymorphisms in immune-regulatory genes such as CD19, CTLA4 and LAG3 have been reported to affect immune checkpoint signaling and treatment response in hematologic malignancies [18–20]. Furthermore, persistence of CAR-T cells and immune- exhaustion patterns correlate with immune-gene profiles and clinical outcome in B-cell lymphomas [21]. These findings indicate that germline variation in immune-related genes may affect immune modulation and therapy responses.” How does the line on CAR-T therapy add to the context of germline variations in genes related to immune functions and therapy response? The study that is being cited correlated clinical outcomes with CAR-T cell persistence in patients with B cell lymphomas. Additionally, what specific immune-gene profiles correlate with CAR-T persistence and immune exhaustion? The statement and context of this sentence in unclear.
  • In Figure 1A, B, C, and D, the authors use the same symbols HW, HT, and HR for both LAG3 and CTLA4 but describe the alleles as I455 or I455T, which is only applicable to LAG3 and not to CTLA4. Why do the authors use so many abbreviations to describe the same variants. In the text, the authors specify T17hom, T17Ahet, and A17hom for CTLA4 and I455hom, I455Thet, T455hom for LAG3. They use these identifiers in panels E and F. They should use the same identifiers in the text and the figures/Tables for better presentation and easier comprehension of the data.
  • The authors must also cite individual Figure panels in the text, where relevant. So, Figure 1A, B, C, D, E, and F must be cited in the text. This comment is applicable to all Figures.
  • The legends and the Figures are mismatched. Figure 1A and B described CTLA4 and LAG3 allele frequencies in European populations, while Figure 1C and D described allele frequencies in patients with multiple myeloma.
  • Within the European population, did the authors note any differences in LAG3 allele frequency distribution between males and females? What was the male to female ratio in the European population? It would be useful to look at the prevalence of these variants in general population by gender.
  • In all genetic models LAG3 rs870849 was significantly associated to risk for multiple myeloma, in the codominant model at allele dosage 2 vs 0 (T455hom vs I455hom) with an OR of 5.4, stratified into male specific risk OR 6.8 and female specific risk OR 3.6 (Figure 3 A,B). The authors should split this sentence to improve readability. The authors should include data for the other genetic models if the risk association was significant. Why is the data from the codominant model alone included?
  • What is VD, sCR/CR, VGPR/PR in Table 2? All abbreviations should be clearly defined in. the legend.
  • The authors should the change phrasing to read associated with instead of associated to. For instance, in this sentence, “In this study the germline variant LAG3 rs870849 was associated to MM risk and to favorable treatment outcome after HDCT/ASCT.”, the correct phrasing is “LAG3 rs870849 was associated wtih MM risk”. The authors should check and make the change where applicable.
  • The authors state in the Discussion that “The effects of LAG3 rs870849 were dose-dependent in both functions of promoting MM development and improving clinical outcome after HDCT/ASCT”, but there is no data to indicate that this risk allele promoted MM development. The odds ratio shows an increased risk for multiple myeloma but there is no clinical data to show that this risk allele promoted development of myeloma or improved clinical outcomes. The comparisons for response to HDCT/ASCT were not statistically significant. The authors should clarify this and amend this statement to reflect the data presented in the manuscript.
  • Another limitation of this study is that it is a single-center study and should be addressed in the Discussion.

Author Response

Reviewer 2

cimb-4029985

Comments and Suggestions for Authors

In this study by Seipel et al., the authors conduct a retrospective analysis of allele frequency of the LAG3rs870849 variant and its association with gender and clinical outcomes in patients with multiple myeloma. The authors show that the homozygous LAG3 rs870849 variant is predominant in multiple myeloma patients and is also more prevalent in male patients than in female patients. Interestingly, patients with the homozygous LAG3rs870849 variant had significantly longer median overall survival.

The manuscript is well written and observations from the study emphasize the need for better characterization of the molecular mechanisms involved in LAG3 rs870849 variant modulation of T-cell activity in different contexts that might contribute to disease risk vs. response to treatment.

Response: We thank the reviewer for the helpful comments. We integrated the responses into the article and believe that the manuscript has been improved.

There are some concerns that the authors must address.

  • It is better to avoid abbreviations in the Abstract. All abbreviations must be expanded in the first instance in the text.

Response: We have modified the abstract.

  • “Genetic polymorphisms in immune-regulatory genes such as CD19, CTLA4 and LAG3 have been reported to affect immune checkpoint signaling and treatment response in hematologic malignancies [18–20]. Furthermore, persistence of CAR-T cells and immune- exhaustion patterns correlate with immune-gene profiles and clinical outcome in B-cell lymphomas [21]. These findings indicate that germline variation in immune-related genes may affect immune modulation and therapy responses.” How does the line on CAR-T therapy add to the context of germline variations in genes related to immune functions and therapy response? The study that is being cited correlated clinical outcomes with CAR-T cell persistence in patients with B cell lymphomas. Additionally, what specific immune-gene profiles correlate with CAR-T persistence and immune exhaustion? The statement and context of this sentence in unclear.

Response. We have removed the sentence and reference.

  • In Figure 1A, B, C, and D, the authors use the same symbols HW, HT, and HR for both LAG3 and CTLA4 but describe the alleles as I455 or I455T, which is only applicable to LAG3 and not to CTLA4. Why do the authors use so many abbreviations to describe the same variants. In the text, the authors specify T17hom, T17Ahet, and A17hom for CTLA4 and I455hom, I455Thet, T455hom for LAG3. They use these identifiers in panels E and F. They should use the same identifiers in the text and the figures/Tables for better presentation and easier comprehension of the data.

Response: We have addressed the definitions and consistency of abbreviations.

  • The authors must also cite individual Figure panels in the text, where relevant. So, Figure 1A, B, C, D, E, and F must be cited in the text. This comment is applicable to all Figures.

Response: We have reorganized the text and cited individual figure panels.

  • The legends and the Figures are mismatched. Figure 1A and B described CTLA4 and LAG3 allele frequencies in European populations, while Figure 1C and D described allele frequencies in patients with multiple myeloma.

Response: We have rearranged the figures.

  • Within the European population, did the authors note any differences in LAG3 allele frequency distribution between males and females? What was the male to female ratio in the European population? It would be useful to look at the prevalence of these variants in general population by gender.

Response: The frequency distribution of autosomal gene variants including LAG3 rs870849 and CTLA4 rs231775 is independent of gender. The male to female ratio in the central European population is 0.99.

  • In all genetic models LAG3 rs870849 was significantly associated to risk for multiple myeloma, in the codominant model at allele dosage 2 vs 0 (T455hom vs I455hom) with an OR of 5.4, stratified into male specific risk OR 6.8 and female specific risk OR 3.6 (Figure 3 A,B). The authors should split this sentence to improve readability. The authors should include data for the other genetic models if the risk association was significant. Why is the data from the codominant model alone included?

Response: As LAG3 proteins dimerize to bind to TCR, the codominant model appeared to be most appropriate. LAG3 protein variants may form heterodimers and all combinatorial effects are possible including dominant, codominant and recessive dimer function. We have added the analysis with all genetic models.

  • What is VD, sCR/CR, VGPR/PR in Table 2? All abbreviations should be clearly defined in. the legend.

Response: Abbreviations have been defined.

  • The authors should the change phrasing to read associated with instead of associated to. For instance, in this sentence, “In this study the germline variant LAG3 rs870849 was associated to MM risk and to favorable treatment outcome after HDCT/ASCT.”, the correct phrasing is “LAG3 rs870849 was associated wtih MM risk”. The authors should check and make the change where applicable.

Response: Associated to has been changed to associated with throughout the text.

  • The authors state in the Discussion that “The effects of LAG3 rs870849 were dose-dependent in both functions of promoting MM development and improving clinical outcome after HDCT/ASCT”, but there is no data to indicate that this risk allele promoted MM development. The odds ratio shows an increased risk for multiple myeloma but there is no clinical data to show that this risk allele promoted development of myeloma or improved clinical outcomes. The comparisons for response to HDCT/ASCT were not statistically significant. The authors should clarify this and amend this statement to reflect the data presented in the manuscript.

Response: We have rephrased the text to clarify the difference of gene risk analysis with risk for multiple myeloma.

  • Another limitation of this study is that it is a single-center study and should be addressed in the Discussion.

Response: We have added this limitation to the discussion. Limitations of the study include the cohort size and the cohort composition with a large majority of patients who underwent HDCT/ASCT (90%). The LAG3 rs870849 allele frequencies were found to be elevated in MM patients eligible for ASCT, but may be lower in MM patients not eligible for ASCT. This may represent a survivorship bias with the implication that LAG3 rs870849 may confer a survival advantage.

Submission Date 21 November 2025

Date of this review 10 Dec 2025 08:02:42

Date of revision: 15 December 2025.

Round 2

Reviewer 1 Report

Comments and Suggestions for Authors

no further comments

Author Response

We thank the reviewer for the helpful comments in the first revision. 

Reviewer 2 Report

Comments and Suggestions for Authors

Most of the comments have been addressed. There are two minor concerns that remain. 

1) There are undefined abbreviations in the Abstract: HDCT/ASCT, MAF, OR. For terms that appear only once, abbreviated forms need not be included (PFS/OS). 

2) The legends and panel labels are still mismatched for Figure 1. The authors should ensure that the Figure panels are cited correctly for Figure 1 in the text. 

Author Response

Cimb-4029985 reviewer 2, round 2

Comments and Suggestions for Authors

Most of the comments have been addressed. There are two minor concerns that remain.

Response: We thank the reviewer for insisting on the minor concerns which had not been fully addressed in the first revision.

  • There are undefined abbreviations in the Abstract: HDCT/ASCT, MAF, OR. For terms that appear only once, abbreviated forms need not be included (PFS/OS). 

Response: All abbreviations have been removed from the abstract.

2) The legends and panel labels are still mismatched for Figure 1. The authors should ensure that the Figure panels are cited correctly for Figure 1 in the text. 

Response: Legends and panel labels in Figure 1 have been matched and cited accordingly.

Submission Date 21 November 2025

Date of this review n16 Dec 2025 20:14:57

Date of revision December 17, 2025